# Neck Circumference and Blood Pressure Measurements among Walter Sisulu University Students

**DOI:** 10.3390/ijerph192215235

**Published:** 2022-11-18

**Authors:** Nthai E. Ramoshaba, Mthetho Q. Fihla, Wenzile S. Mthethwa, Lisa Tshangela, Zuqaqambe M. Mampofu

**Affiliations:** Department of Human Biology, Walter Sisulu University, Nelson Mandela Drive, Mthatha 5117, South Africa

**Keywords:** neck circumference, blood pressure, obesity, students, subcutaneous fat

## Abstract

Hypertension is a public health problem in South Africa. Increases in subcutaneous fat, presented by the neck circumference (NC) value, contribute to and predict the development of hypertension. However, to date, there has been no study done to investigate the relationship between the NC and blood pressure (BP) among historically disadvantaged university students. Therefore, the aim of the study was to investigate the relationship between the NC and BP among Walter Sisulu University students. This cross-sectional study was conducted in Walter Sisulu University. All 127 students were aged 18 years and above and underwent NC and clinical BP measurements using standard procedures. In a Pearson’s correlation analysis, the NC positively correlated with both the systolic blood pressure (SBP) (r = 0.5; *p* < 0.001) and diastolic blood pressure (DBP) (r = 0.3; *p* < 0.001). Furthermore, in the multivariable-adjusted regression analysis, the NC was positively associated with both the SBP (adjusted R^2^ = 0.3, β = 2.0 (95% CI = 1.1; 2.9), *p* < 0.001) and DBP (adjusted R^2^ = 0.1 β = 0.950 (95% CI = 0.3; 1.6), *p* = 0.008) adjusted for age, gender, body mass index, waist-to-height ratio, alcohol, and smoking. The NC is related to BP among historically disadvantaged university students.

## 1. Introduction

Hypertension or raised blood pressure (BP) poses a serious public health threat, which contributes significantly to the illness burden in low- and middle-income countries, including South Africa [1,2,3]. It is estimated that 46% of people with hypertension are unaware that they have this condition, and by the year 2025 it is expected that more than 1.56 billion people worldwide will suffer from hypertension [2,4]. In addition, central obesity is one of the major risk factors of morbidity and mortality in hypertension [2,5,6].

Several anthropometric measurements such as the mid-upper arm circumference, waist circumference (WC), waist-to-height ratio, and neck circumference (NC) are widely employed as indicators of central obesity [7,8,9,10,11]. These circumferences that measure different fat distributions on the body have been positively associated with BP [10,12,13,14,15].

The NC measures the subcutaneous fat of the upper body and has been reported to be a valuable tool that is cheap and simple to use and can result in time savings [16,17,18]. In addition, studies have shown that the NC is closely related to the development of metabolic disorders and hypertension [19,20]. At present, there is no study on the relationship between the NC and BP results among historically disadvantaged university students. Therefore, it is crucial to determine the relationship between NC and BP among such students in order to predict the development of hypertension, which will help in the early prevention of hypertension. This current study investigated the correlation between the NC and BP among Walter Sisulu University students.

## 2. Materials and Methods

### 2.1. Geographical Area

Walter Sisulu University is one of the historically disadvantaged universities in South Africa, located in the Eastern Cape Province. The ethnic composition of Walter Sisulu University students is approximately 97% Black, 2% Indian, and 1% White.

### 2.2. Study Design

This study employed a cross-sectional study design, with a sample size of 127 Black students aged 18 and above from Walter Sisulu University, Mthatha Campus, Nelson Mandela Drive.

### 2.3. Ethical Consideration

This study was conducted according to the guidelines of the Declaration of Helsinki and was approved by the Human Ethics Committee of Walter Sisulu University (protocol number: 096/2020). A written informed consent form was given to each student prior data collection.

### 2.4. Recruitment

Students were recruited by word of mouth from their residences to the physiology laboratory where the data were collected.

### 2.5. Questionnaire Data

A general demographic and lifestyle questionnaire was completed by each student and the data with regards to age, sex, self-reported smoking (yes/no), and alcohol consumption (yes/no) were collected.

### 2.6. Anthropometric Measurements

The guidelines of the International Society for the Advancement of Kinanthropometry were followed for all anthropometric measurements [21]. The NC and WC were measured to the nearest 0.1 cm using a flexible steel tape (Lufkin Steel Tape; W606PM; Lufkin, TX, USA; Apex, NC, USA). The NC was measured perpendicular to the long axis of the neck, directly above the thyroid cartilage (the Adam’s apple). The students were standing during the measurement but kept their head in the Frankfort plane. Since the tissues in this area are compressible, we avoided pulling the tape too tightly when measuring. 

The WC measurements were taken at the level of the narrowest point between the iliac crest and the bottom part of the thoracic cage, with the students standing in an upright position and after mild expiration. The students assumed a relaxed standing position with the arms folded across the thorax.

The body height was measured to the nearest 0.1 cm using a SECA 213 Portable Stadiometer (SECA, Hamburg, Germany). For body height measurements, the students had to stand with their feet together and their heels, buttocks, and upper back touching the scale. The students were asked to take and hold a deep breath while keeping their head in the Frankfort plane. We applied a gentle upward lift through the mastoid processes. The base of the stadiometer was then lowered to the vertex of the head, and if there was a lot of hair on head a little pressure was applied to touch the top of the head.

The body weight was measured to the nearest 0.1 kg using an electronic scale (SECA, Hamburg, Germany). Before the student climbed onto the scale, we checked if the scale reading was zero, then the student stood at the center of the scale without support and with their weight distributed evenly on both feet. The head was positioned upwards, with the eyes looking directly ahead. Both the body mass index (weight (kg)/height (m)^2^) and waist-to-height ratio (WC (cm)/height (cm)) were calculated.

#### Quality Control

All training of anthropometric measurements was done in accordance with the International Society for the Advancement of Kinanthropometry [21]. The absolute and relative values for intra- and inter-evaluator technical error of measurements’ (TEM %) values for NC were 0.27% and 0.31%, respectively; the body height values were 0.31 and 0.42%, respectively; the body weight values were 0.10 and 0.21, respectively; and the WC values were 0.33 and 0.46%, respectively.

### 2.7. Clinical Blood Pressure Measurements

The clinical BP was measured using an Omron M3 BP monitor (Omron, Kyoto, Japan). Three readings of the systolic BP (SBP), diastolic BP (DBP), and heart rate were taken at intervals of five minutes after the students had been seated for five minutes or longer. The averages of three readings were recorded [22]. The pulse pressure (PP) was derived by subtracting the systolic BP from the diastolic BP. The BP values were categorized into elevated BP (SBP = 120–129 mmHg, DBP < 80 mmHg), prehypertension (SBP = 130–139, DBP = 80–89 mmHg), and hypertension (SBP ≥ 140 mmHg or DBP ≥ 90 mmHg) [23].

### 2.8. Statistical Analysis

The Kolmogorov–Smirnov test as well as graphical methods (histograms and q-q plots) were used to assess the normality. The normally distributed continuous data were presented as the arithmetic mean and standard deviation. The independent t-test analysis was applied to test the differences in continuous variables between males and females. The categorical data (gender, alcohol status, and smoking status) were presented as frequencies and proportions. Chi-square tests (categorical variables) were used to test differences between male and female students. A Pearson correlation analysis was performed to determine the relationship between the NC and BP. A multivariate regression analysis was performed to investigate associations between the NC and BP. adjusted for age, gender, body mass index, waist-to-height ratio, alcohol, and smoking. All statistical analyses were performed using the statistical package for the social sciences (SPSS) version 26 (SPSS, Chicago, IL, USA). The statistical significance was set at *p* < 0.05. A power calculation revealed that a minimum sample size of *n* = 89 would be required to perform our multivariate regression analysis with an effect size of 0.15, alpha set to 0.05, and power to 0.95.

## 3. Results

Of the total group of students, the means of the NC, BMI, WHtR, SBP, DBP, and PP were 32.6 cm, 24.4 kg/m^2^, 0.5, 107.8 mmHg, 68.7 mmHg, and 39.1 mmHg, respectively (Table 1). The prevalence of smoking was 13.4%, while the alcohol consumption rate was 33.9% among the students. The percentages of students with elevated blood pressure and prehypertension equaled 11% and 5.5%, respectively. Hypertension was not present among the students. After comparing the characteristics between male and female students, the male students had a bigger mean NC (34.8 vs. 31.6 cm, *p* < 0.001) than the female students. The male students had a higher SBP (112.3 vs. 105.6 mmHg, *p* = 0.002) than the female students, while no DBP differences were observed between male and female students. There were no significant differences in BP categories between male and female students.

In the Pearson correlation analysis, the NC positively correlated with the SBP (r = 0.5; *p* < 0.001), DBP (r = 0.3; *p* = 0.001), and PP (r = 0.4; *p* < 0.001) (Table 2). 

Table 3 shows the multivariable-adjusted regression analysis results, whereby the NC was positively associated with the SBP (adjusted R^2^ = 0.252, β = 2.001 (95% CI = 1.1; 2.9), *p* < 0.001), DBP (adjusted R^2^ = 0.096 β = 0.950 (95% CI = 0.3; 1.6), *p* = 0.008), and PP (adjusted R^2^ = 0.162 β = 1.1 (95% CI = 0.4; 1.7), *p* < 0.001) adjusted for age, gender, body mass index, waist-to-height ratio, alcohol, and smoking.

## 4. Discussion

This cross-sectional study investigated the relationship between the NC and BP among historically disadvantaged university students. We found for the first time a positive association between the NC and BP among students adjusted for age, gender, body mass index, waist-to-height ratio, alcohol, and smoking.

Our results agree with previous studies that reported a positive association between the NC and BP in children and adults from developing countries [7,15,18,24,25,26]. In a systematic review and meta-analysis, the NC was strongly associated with the BP in children [15]. In addition, Ben-Noun and Laor [7] reported that the NC strongly correlated, positively with the BP in adults with no hypertension. Recently, it was observed that the NC was associated positively with hypertension in adults [18,24]. These previous findings, together with our current findings on the positive relationships between the NC and BP, indicate that the NC is independently associated with an increase in BP. Our results suggest that already in young students, an increase in NC may contribute to elevated BP [15], a likely precursor or predictor of the development of cardiovascular diseases. This has to be further explored in longitudinal studies to clarify our findings.

The NC is better in predicting obesity-related health risks than the body mass index, as the body mass index does not differentiate between fat and other tissues such as bones and muscles [27]. In addition, the body mass index cannot account for the regional fat distribution, which was reported to be more pathogenic than the total adiposity [28]. The NC measures subcutaneous fat in the upper body and is a valuable tool to indicate visceral obesity [16,17]. An increase in subcutaneous fat stimulates the release of free fatty acids into the blood stream [29,30]. The increase in blood fatty acids leads to increased inflammatory risk factors and oxidative stress by increasing the production of oxygen free radicals and decreasing antioxidant concentrations, which subsequently results in insulin resistance and vascular damage [31,32,33]. Therefore, both insulin resistance and vascular deterioration have been associated with increased BP [33,34].

The mechanism that links the NC and BP is not clear yet; however, it may be explained by the following points. High adiposity as indicated by an increase in NC results in the synthesis and release of leptin. Leptin will then activate the sympathetic nervous system, which triggers the secretion of norepinephrine and epinephrine, which will cause an increase in heart rate and vascular constriction [35,36], subsequently leading to elevated BP. On the other hand, obesity may lead to high levels of F2-isoprostaglandin, which will cause vascular endothelial damage [37,38]. All proposed mechanisms will ultimately lead to the development of hypertension. Therefore, tracing NC changes could assist in identifying individuals who are at risk of developing hypertension, which was reported to be a public health problem in South Africa [3].

Our findings should be interpreted in the context of some limitations. The current study had a cross-sectional design; thus, the causality cannot be inferred. We did not consider other confounders or covariates such as the family history, food consumption, physical activity, and stress factors in this study. However, we recommend future studies to add these confounders to the linear regression models to confirm our findings among the students. The data were collected only in one visit, and we further reported a low percentage of prehypertension, which may have been because the students were young in this study. Therefore, longitudinal studies on the relationship between the NC and BP are needed to track the NC changes with BP over time. The NC and BP are more advantageous measurement tools, due to their applicability and being easy to understand by the general population. 

## 5. Conclusions

In a historically disadvantageous university, the NC was positively related to the BP among the students. The NC is a simple and useful tool to identify individuals who are at risk of developing hypertension. 

## Figures and Tables

**Table 1 ijerph-19-15235-t001:** Characteristics of Walter Sisulu University students.

Characteristics	Total Group (N = 127)	Female (N = 86)	Male (N = 41)	*p*-Value
Age (Years)	20.9 ± 2.4	20.2 ± 2.0	22.3 ± 2.5	<0.001
NC (cm)	32.6 ± 2.7	31.6 ± 1.9	34.8 ± 2.9	<0.001
BMI (kg/m^2^)	24.4 ± 4.2	24.2 ± 3.8	24.4 ± 5.0	0.800
WHtR	0.5 ± 0.1	0.5 ± 0.1	0.5 ± 0.1	0.369
SBP (mmHg)	107.8 ± 11.3	105.6 ± 10.9	112.3 ± 11.0	0.002
DBP (mmHg)	68.7 ± 8.2	68.2 ± 8.2	69.6 ± 8.0	0.360
HR (bpm)	78.0 ± 12.1	80.4 ± 12.0	73.1 ± 10.9	0.001
PP (mmHg)	39.1 ± 7.9	37.4 ± 7.4	42.7 ± 7.7	<0.001
Elevated BP, n (%)	14 (11.0%)	6 (7.0%)	8 (19.5%)	0.079
Prehypertension, n (%)	7 (5.5%)	4 (4.7%)	3 (7.3%)	
Smoking, n (%)	17 (13.4%)	5 (5.8%)	12 (29.3%)	<0.001
Alcohol, n (%)	43 (33.9%)	14 (16.3%)	29 (70.7%)	<0.001

Abbreviations: NC, neck circumference; BMI, body mass index; WHtR, waist-to-height ratio; SBP, systolic blood pressure; DBP, diastolic blood pressure; HR, heart rate; PP, pulse pressure; BP, blood pressure.

**Table 2 ijerph-19-15235-t002:** Pearson correlations between the neck circumference and blood pressure.

	Neck Circumference (cm)
SBP (mmHg)	r = 0.5; *p* < 0.001
DBP (mmHg)	r = 0.3; *p* = 0.001
PP (mmHg)	r = 0.4; *p* < 0.001

Abbreviations: SBP, systolic blood pressure; DBP, diastolic blood pressure; PP, pulse pressure.

**Table 3 ijerph-19-15235-t003:** Independent associations between the BP as the dependent variable and the NC as the main independent variable.

	SBP (mmHg)		DBP (mmHg)		PP (mmHg)	
	Adjusted R^2^ = 0.252		Adjusted R^2^ = 0.096		Adjusted R^2^ = 0.162	
Independent variables	β (95% CI)	*p*-value	β (95% CI)	*p*-value	β (95% CI)	*p*-value
NC (cm)	2.0 (1.1; 2.9)	<0.001	1.0 (0.3; 1.6)	0.008	1.1 (0.4; 1.7)	<0.001
Age (years)	−0.1 (1; 0.8)	0.848	−0.1 (−0.7; 0.7)	0.971	−0.1 (−0.7; 0.6)	0.825
Women/men, n (%)	−1.7 (−6.9; 3.5)	0.520	1.8 (−2.3; 5.9)	0.386	−3.5 (−7.4; 0.3)	0.072
BMI (kg/m^2^)	1.1 (−0.1; 2.00)	0.023	0.9 (0.1; 1.6)	0.018	0.2 (−0.5; 0.9)	0.588
WHtR	−58 (−118.6; 2.1)	0.058	−46 (−94.7; 1,0)	0.055	−11.4 (−55.8; 33.1)	0.614
Smoking, n (%)	3.5 (−2.3; 9.3)	0.232	2.2 (−2.3; 6.8)	0.338	1.3 (−1.0; 5.5)	0.551
Alcohol, n (%)	−3.0 (−7.8; 1.7)	0.206	0.0 (−3.7; 3.8)	0.982	−3.1 (−6.6; 0.4)	0.082

Abbreviations: NC, neck circumference; BMI, body mass index; WHtR, waist-to-height ratio; SBP, systolic blood pressure; DBP, diastolic blood pressure; PP, pulse pressure.

## Data Availability

The data presented in this study are available on request from the corresponding author.

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
