# Peer review of "Neck Circumference and Blood Pressure Measurements among Walter Sisulu University Students"

_ijerph, 2022, doi:10.3390/ijerph192215235_

Round 1

Reviewer 1 Report

Dear authors, 

The paper is interesting yet extensive revision is needed especially in the introduction and methodology sections. 

The introduction is very brief and does not provide enough background information. 

As for the methodology, it needs much more detailing and elaboration in all sections,, as it does not provide any information on the procedure or the validity or sample calculation or response rate etc... Massive revision should be done in the methods to be acceptable. 

Best.

Author Response

Dear Reviewer 1 

General response: Thank you for the comments and suggestions. Indeed, after addressing the comments below, the quality of the manuscripts has improved.

Comment 1: The introduction is very brief and does not provide enough background information. 

Response 1: Thank you. The introduction was updated. Please see page 1, line 26 -29:

It is estimated that 46% of people with hypertension are unware that they have this condition and by the year 2025, it is expected that more than 1.56 billion people worldwide would suffer from hypertension [2, 4]. In addition, central obesity is one of the major risk factors of morbidity and mortality of hypertension (page 1, line 26-29).

“NC is closely related to the development of metabolic disorder and hypertension” (Page 1, line 36)

Comment 2: As for the methodology, it needs much more detailing and elaboration in all sections,, as it does not provide any information on the procedure or the validity or sample calculation or response rate etc... Massive revision should be done in the methods to be acceptable. 

Respond 2 a: Thank you. The method section was elaborated in details and we also cited the guidelines that we followed for all the measurements in this study (International Society for the Advancement of Kinanthropometry were followed for all anthropometric measurements; American Heart Association Hypertension Guideline was  followed for BP measurements). Please see the methodology section (We highlighted the information added)

Respond 2 b.  Thank you. About the sample size, we used G Power analysis and it revealed that minimum sample size of 89 is required to perform linear regressions. Please see page 3, line 124-126:

A power calculation revealed that minimum sample size of N = 89 would be required to perform our multivariate regression analysis with parameters of effect size of 0.15, alpha set at 0.05 and power at 0.95

Respond 2c: Thank you. About quality control. We calculated intra-evaluator and enter-evaluator test error of measurements (% TEM). Our % TEM are acceptable. Please see Page 3, line 98-103:

2.6.1 Quality control

All training of anthropometric measurements were done in accordance with the International Society for the Advancement of Kinanthropometry [20]. The absolute and relative values for intra- and inter-evaluator technical error of measurements (TEM %) for NC was 0.27 % and 0.31 % respectively; body height was 0.31 and 0.42 % respectively; body weight was 0.10 and 0.21 respectively; WC was 0.33 and 0.46 % respectively

Reviewer 2 Report

  The current paper under the title “Neck circumference and blood pressure among Walter Sisulu University Students” is presented by Nthai E Ramoshaba, Mteto Q Fihla,  Wenzile S Mthethwa, Lisa Tshangela, Zuqaqambe M Mampofu from the Department of Human Biology at Walter Sisulu University in South Africa.  

  Authors present an analysis on data from a cross sectional study including 127 students aged 18 years and above who underwent anthropometric measurements and blood pressure measurements. Authors aimed to assess “the correlation between NC and BP among Walter Sisulu University students” after adjusting for age, gender, body mass index, smoking habit, and alcohol intake.

  Overall, the study is well presented and somehow interesting. Sections are clearly introduced and described, and the statistical analysis is correct but insufficient. In this sense, this study has several flaws and issues that should be addressed by authors. The main question that arises is the validity of this analysis on a young university subjects’ population where none of the recruited students is hypertensive. This issue is critical, as many factors influence blood pressure (BP) levels in these students, such as regular practice of sports, daily salt, vegetables and whole grains intake, control of stress and family history, none of them have been evaluated in this study. In my opinion, the best approach to study the relationship between neck circumference (NC) and BP would be if NC is related to the diagnosis of hypertension, but this will require hypertensive subjects which are quite infrequent at this age. To avoid this problem, authors could follow more than one way. First, to perform analysis with different BP thresholds such as 130/80 mm Hg, as grade I Hypertension stage has been defined at that threshold by the American Heart Association. Moreover, according to the American Heart Association, normal BP for adults (ages 20 and older) is less than 120/80 mm Hg, which could be also selected as a threshold to perform and analyse. Second, if no or very few subjects are found with those thresholds, it could have been interesting to follow-up this subjects and study if the BP measurements remained higher, although not reaching the selected thresholds, after several months or years. In fact, when diagnosing high BP, two or more BP readings at separate medical appointments are recommended by societies.

 Another important fact not addressed by authors is that of gender and ethnicity. When researchers for the National Center for Health Statistics looked at average blood pressure in U.S. adults between 2001 and 2008, the average reading was 122/71 mm Hg. The breakout was 124/72 mm Hg for men, and 121/70 mm Hg for women. It rose by age and was significantly higher in Black people (Wright JD, et al. Natl Health Stat Report. 2011 Mar;(35):1-22, 24). Thus, also a separate analysis should be carried out according to sex, and authors should specify which is the ethnicity or ethnic groups of the included subjects from Walter Sisulu University.

 Regarding alcohol intake and smoking habits, they have not been defined. I guess smoking has been defined at least one cigarette every day. I guess that there were not past smokers (those who quit more than 1 year ago). Regarding alcohol intake, it is quite important its definition. In University population most of alcohol intake would probably be at weekends, thus BP could be influenced by the day subjects have been phenotyped, such as mondays. The same for smoking habit, as this population is more prone to smoke more cigarettes at leisure days, with previous evidence that smoking increases BP acutely.

  On the other hand, the BMI even though is the most used measure to diagnose obesity, it has increasingly been questioned as a surrogate marker for diagnosis of body fat, particularly visceral fat. In this sense, I would recommend authors to build regression analysis introducing other anthropometric indices, such as waist to height or waist to hip ratio, to verify if BP levels are still higher in subjects with higher NC beyond those body fat anthropometric indices.

  Discussion and Limitations sections are too poor. Authors should include a more in-depth comparison with previous studies, indicating if the relationship found in previous young or general population studies has been weak or strong, or if there exists inconclusive reported data. Regarding limitations, BP measurements have been taken only in one visit, this limitation should be acknowledged. In this sense, another measure in a different day would have increased the reliability of the results. As previously stated, practice of sports, daily salt, vegetables and whole grains intake, control of stress and family history are confounding factors that have not been evaluated in this study and could act as confounding factors.

  Minor comments 

  • If pulse pressure has been determined as stated in the Methods section, it should be included later in the characteristics table and in the analyses.
  • Table 2 should be better a figure with “r” results in graphics.
  • Table 3. The only important results to show here at the NC β results. The other β results are meaningless neither have significant p-values.

  Authors should correct following minor grammar or syntax errors such as: 

o   Abstract: “All 127 students who aged 18 years and above, underwent NC and clinic BP measurements using standard procedure.”

o   Discussion section: “The mechanism that links the NC and BP is not clear yet, however, it may be explained from the following points. Increased adiposity as indicated by increase NC, results in synthesis and release of leptin that will increase the sympathetic nervous system activity, which will secrete norepinephrine and epinephrine that will cause an increase in heart rate and vascular constriction [27, 28]. Subsequently leading to increase in BP. On the other hand, increases in obesity may lead to an increase in F2-isoprostaglandin which then will lead to vascular endothelial damage [29, 30]”. I suggest rewriting as the term “increase” has been repeated too many times.

Author Response

Dear Reviewer 2

General comments 1: Overall, the study is well presented and somehow interesting. Sections are clearly introduced and described, and the statistical analysis is correct but insufficient. In this sense, this study has several flaws and issues that should be addressed by authors.

Response 1: Thank you for the comments and suggestions. Indeed, after addressing the comments below, the quality of the manuscripts has improved.

Comments 2: The main question that arises is the validity of this analysis on a young university subjects’ population where none of the recruited students is hypertensive. This issue is critical, as many factors influence blood pressure (BP) levels in these students, such as family history, food consumption, physical activity and stress factors

Response 2: Thank you. We agree that many factors may influence BP levels  such as regular practice of sports, daily salt, vegetables and whole grains intake, control of stress and family history. However, in this study we did not consider those factors. We included them as part of limitation of the study. Please see page 5, line 193-195:

In this current study, we did not consider other confounders or covariates such as family history, food consumption, physical activity and stress factors. However, we recommend future studies to add these confounders in the linear regression models

Comment 3 :  In my opinion, the best approach to study the relationship between neck circumference (NC) and BP would be if NC is related to the diagnosis of hypertension, but this will require hypertensive subjects which are quite infrequent at this age. To avoid this problem, authors could follow more than one way. First, to perform analysis with different BP thresholds such as 130/80 mm Hg, as grade I Hypertension stage has been defined at that threshold by the American Heart Association. Moreover, according to the American Heart Association, normal BP for adults (ages 20 and older) is less than 120/80 mm Hg, which could be also selected as a threshold to perform and analyse.

Response 3: Thank you. In the methodology section, we did categorise BP levels in to to elevated BP, stage 1 hypertension and stage 2 hypertension following American College of Cardiology/American Heart Association Hypertension Guideline [22]. The results were reported in Table 1.  Please see, page 3, line 109-111 and page 4 line 136-137 (Table 1):

 The BP was categorized in to elevated BP (SBP = 120-129 mmHg, DBP < 80 mmHg); stage 1 hypertension (SBP = 130-129, DBP = 80-89 mmHg); stage 2 hypertension ( SBP ≥ 140 mmHg or DBP ≥ 90 mmHg)[22] (Please see page 3, line 109-111)

The percentages of students with elevated blood pressure and stage 1 hypertension were 11% and 5.5% respectively. Stage 2 hypertension was not present among the students. (Page 4 line 136-137)

Comment 4: Second, if no or very few subjects are found with those thresholds, it could have been interesting to follow-up this subjects and study if the BP measurements remained higher, although not reaching the selected thresholds, after several months or years. In fact, when diagnosing high BP, two or more BP readings at separate medical appointments are recommended by societies.

Response 4: Thank you. We found low percentage of stage 1 hypertension (see response 3 above). With regards to two or more visit for BP measurements, In this study we measured blood pressure in one visit only. However, we included as part of limitation of the study. Please see Page 5, line 195 -197:

The data was collected only in one visit and we reported low percentage of stage 1 hypertension. Therefore, longitudinal studies are needed to track the NC changes with BP over time.

 Comment 5: Another important fact not addressed by authors is that of gender and ethnicity. When researchers for the National Center for Health Statistics looked at average blood pressure in U.S. adults between 2001 and 2008, the average reading was 122/71 mm Hg. The breakout was 124/72 mm Hg for men, and 121/70 mm Hg for women. It rose by age and was significantly higher in Black people (Wright JD, et al. Natl Health Stat Report. 2011 Mar;(35):1-22, 24). Thus, also a separate analysis should be carried out according to sex, and authors should specify which is the ethnicity or ethnic groups of the included subjects from Walter Sisulu University.

Response 5: Thank you. The ethnic composition of Walter Sisulu University is approximately:  97% Blacks, 2% Indians and 1% Whites ( Page 2, line 48 -49). This study composed of black students only (Page 2, line 51 to 53). For gender , we did compare the characteristic between the male and female students (Table 1)

The ethnic composition of Walter Sisulu University students is approximately: 97% Blacks, 2% Indians and 1% Whites (Page 2, line 48 -49).

This study employed a cross-sectional study design, with a sample size of 127 Black students aged 18 and above from Walter Sisulu University, Mthatha campus, Nelson Mandela Drive (Page 2, line 51 to 53)

Comment 6: Regarding alcohol intake and smoking habits, they have not been defined. I guess smoking has been defined at least one cigarette every day. I guess that there were not past smokers (those who quit more than 1 year ago). Regarding alcohol intake, it is quite important its definition. In University population most of alcohol intake would probably be at weekends, thus BP could be influenced by the day subjects have been phenotyped, such as mondays. The same for smoking habit, as this population is more prone to smoke more cigarettes at leisure days, with previous evidence that smoking increases BP acutely.

Response 6: Thank you. For both alcohol and smoking, it was current self-report, yes or no. Please see the method section (page 2 65 to 66):

  “self-reported smoking (yes/no) and alcohol consumption (yes/no) was collected”.

 Comment 7:  On the other hand, the BMI even though is the most used measure to diagnose obesity, it has increasingly been questioned as a surrogate marker for diagnosis of body fat, particularly visceral fat. In this sense, I would recommend authors to build regression analysis introducing other anthropometric indices, such as waist to height or waist to hip ratio, to verify if BP levels are still higher in subjects with higher NC beyond those body fat anthropometric indices.

Response 7: Thank you. The waist to height was included in the linear model. The NC remain independently associated with BP. Please see Table 3  Page 4,line 154-157):

NC positively associated with SBP [Adjusted R2 = 0.252, β = 2.001 (95% CI = 1.1; 2.9), P < 0.001], DBP [Adjusted R2 = 0.096 β = 0.950 (95% CI = 0.3; 1.6), P = 0.008] and PP [Adjusted R2= 0.162 β = 1.1 (95% CI = 0.4; 1.7), P < 0.001] adjusted for age, gender, body mass index, waist-to-height ratio, alcohol and smoking.

Comment 8: Discussion and Limitations sections are too poor. Authors should include a more in-depth comparison with previous studies, indicating if the relationship found in previous young or general population studies has been weak or strong, or if there exists inconclusive reported data.

Response 8: Thank you. The comparison was improved in the manuscript. Please see page 5, line 165 -171:

Our results are in agreement with previous studies that reported a positive association between NC and BP in children and adults from developing countries [7, 14, 17, 23-25]. In systematic review and meta-analysis, NC strongly associated with BP in children [14]. In addition, Ben-Noun and Laor [7], reported that NC strongly correlated positively with BP in adults with no hypertension. Recently, it was observed that NC associated positively with hypertension in adults [17, 23]. These previous findings together with our current findings on the positive relationship between NC and BP, indicate that when NC increases, the BP also increases

Comment 9: Regarding limitations, BP measurements have been taken only in one visit, this limitation should be acknowledged. In this sense, another measure in a different day would have increased the reliability of the results. As previously stated, practice of sports, daily salt, vegetables and whole grains intake, control of stress and family history are confounding factors that have not been evaluated in this study and could act as confounding factors.

Response 9: Thank you.  We included them as part of limitation. Please see page 5, line 193 -198:

In this current study, we did not consider other confounders or covariates such as family history, food consumption, physical activity and stress factors. However, we recommend future studies to add these confounders in the linear regression models. The data was collected only in one visit and we reported low percentage of stage 1 hypertension. Therefore, longitudinal studies are needed to track the NC changes with BP over time.

General Minor comments 

  • Comment 1: If pulse pressure has been determined as stated in the Methods section, it should be included later in the characteristics table and in the analyses.

Response 2 : Thank you. The pulse pressure was included in the Tables (1, 2, 3). Please see page 4, line 154 -157:

NC positively associated with SBP [Adjusted R2 = 0.252, β = 2.001 (95% CI = 1.1; 2.9), P < 0.001], DBP [Adjusted R2 = 0.096 β = 0.950 (95% CI = 0.3; 1.6), P = 0.008] and PP [Adjusted R2= 0.162 β = 1.1 (95% CI = 0.4; 1.7), P < 0.001] adjusted for age, gender, body mass index, waist-to-height ratio, alcohol and smoking.

  • Comment 3: Table 2 should be better a figure with “r” results in graphics.

Response 3: Thank you. We felt that the table is good.

  • Comment 4: Table 3. The only important results to show here at the NC β results. The other β results are meaningless neither have significant p-values.

Response 4: Thank you. We thought that is important to show how each covariate contributed in the model in details.  

  Authors should correct following minor grammar or syntax errors such as: 

Comment 5: Abstract: “All 127 students who aged 18 years and above, underwent NC and clinic BP measurements using standard procedure.”

Response 5: Thank you. The sentenced was corrected, Abstract, line 13 -14:

“All 127 students aged 18 years and above, underwent NC and clinic BP measurements using standard procedures”

Comment 6: Discussion section: “The mechanism that links the NC and BP is not clear yet, however, it may be explained from the following points. Increased adiposity as indicated by increase NC, results in synthesis and release of leptin that will increase the sympathetic nervous system activity, which will secrete norepinephrine and epinephrine that will cause an increase in heart rate and vascular constriction [27, 28]. Subsequently leading to increase in BP. On the other hand, increases in obesity may lead to an increase in F2-isoprostaglandin which then will lead to vascular endothelial damage [29, 30]”. I suggest rewriting as the term “increase” has been repeated too many times.

Response 6: Thank you. The paragraph was corrected page 5, line 182 -188:

High adiposity as indicated by an increase in NC, results in the synthesis and release of leptin. Leptin will then activate the sympathetic nervous system, which triggers the secretion of norepinephrine and epinephrine that will cause an increase in heart rate and vascular constriction [32, 33]. Subsequently leading to elevated BP. On the other hand, obesity may lead to high levels of F2-isoprostaglandin which will cause vascular endothelial damage [34, 35]. All these proposed mechanisms will ultimately lead to the development of hypertension

Round 2

Reviewer 1 Report

Dear authors thank you for incorporating the comments. 

Author Response

Dear Reviewer 1, 

Thank you again for reviewing the manuscript.

Reviewer 2 Report

The current paper under the title “Neck circumference and blood pressure among Walter Sisulu University Students” is presented by Nthai E Ramoshaba, Mteto Q Fihla,  Wenzile S Mthethwa, Lisa Tshangela, Zuqaqambe M Mampofu from the Department of Human Biology at Walter Sisulu University in South Africa.  

 Authors present an analysis on data from a cross sectional study including 127 students aged 18 years and above who underwent anthropometric measurements and blood pressure measurements. Authors aimed to assess “the correlation between NC and BP among Walter Sisulu University students” after adjusting for age, gender, body mass index, waist-to-height ratio, smoking habit, and alcohol intake.

  Overall, the study has improved after the authors have added most of the suggestions. Although the study lacks novelty, it is well presented, easy to read and interesting, as authors have described the relationship between neck circumference in black people from SouthAfrica  and higher blood pressure. Sections are clearly introduced and described, and the statistical analysis is correct.

 Minor issues: in the abstract, authors should indicate that in the multivariate regression analysis they also adjusted with waist to height ratio. In the Introduction section, when discussing the several anthropometric circumference measurements widely used, they should also mention the waist to height ratio, which has demonstrated a clear relationship with cardiovascular risk and for diabetes and hypertension, thus it is later used to adjust in the multivariate regression. In the Clinic blood pressure measurements section, I suggest to specify as follows: “The BP was categorized in to prehypertension or elevated BP (SBP = 109 120-129 mmHg, DBP < 80 mmHg);[…]”.

Author Response

Dear Reviewer 2

Thank you again for the comments and suggestions given below. They improved the manuscript.

Comment 1: in the abstract, authors should indicate that in the multivariate regression analysis they also adjusted with waist to height ratio.

Response 1: Thank you, the waist to height ratio was inserted in the abstract:

Furthermore, in multivariable-adjusted regression analysis, NC positively associated with both SBP [Adjusted R2 = 0.3, β = 2.0 (95% CI = 1.1; 2.9), P < 0.001] and DBP [Adjusted R2 = 0.1 β = 0.950 (95% CI = 0.3; 1.6), P = 0.008] adjusted for age, gender, body mass index, “waist-to-height ratio”, alcohol, and smoking

Comment 2: In the Introduction section, when discussing the several anthropometric circumference measurements widely used, they should also mention the waist to height ratio, which has demonstrated a clear relationship with cardiovascular risk and for diabetes and hypertension, thus it is later used to adjust in the multivariate regression.

Response 2: Thank you, the waist to height ratio was included in the introduction and the reference was also added to support the statement. Please see page 1, line 33-35:

Several anthropometric measurements such as mid-upper arm circumference, waist circumference (WC), “waist-to-height ratio” and neck circumference (NC) are widely employed as indicators of central obesity [7-11]

Comment 3: In the Clinic blood pressure measurements section, I suggest to specify as follows: “The BP was categorized in to prehypertension or elevated BP (SBP = 109 120-129 mmHg, DBP < 80 mmHg); […]”.

Response 3: Thank you, the term was changed from stage 1 hypertension to prehypertension as suggested. Then, we used prehypertension throughout the manuscript. Please see page 3, line 112-114

The BP was categorized in to elevated BP (SBP = 120-129 mmHg, DBP < 80 mmHg); “prehypertension”(SBP = 130-139, DBP = 80-89 mmHg); hypertension ( SBP ≥ 140 mmHg or DBP ≥ 90 mmHg)[23].